# Effect of Yogurt and Its Components on the Deodorization of Raw and Fried Garlic Volatiles

**DOI:** 10.3390/molecules28155714

**Published:** 2023-07-28

**Authors:** Manpreet Kaur, Sheryl Barringer

**Affiliations:** Department of Food Science and Technology, The Ohio State University, Columbus, OH 43210, USA; kaur.333@osu.edu

**Keywords:** garlic, cooking, volatiles, deodorization, yogurt, fat, proteins, heating, pH

## Abstract

Garlic contains sulfur volatiles that cause a bad odor after consumption. The objective of this study was to understand how yogurt and its components cause deodorization. Raw and fried garlic samples were mixed with various treatments and measurements of volatiles were conducted using a selected-ion flow-tube mass spectrometer. Frying garlic significantly reduced almost all sulfur volatile compounds. Raw garlic was deodorized more than fried garlic by all of the treatments. Fat, protein and water significantly reduced the concentration of sulfur-based volatiles in garlic. At the same concentration, either fat or protein produced higher deodorization, depending on the hydrophobicity of the volatile. Whey protein, casein and their complex all caused deodorization. Increasing the pH to 7 or heating changed the structure of the proteins and decreased the deodorization of the volatiles, showing the importance of proteins for deodorization. As the quantity of fat increased, the deodorization of the volatiles also increased. Foods with higher fat or protein content can be formulated to offer a potential solution to reduce the unpleasant odor associated with garlic consumption.

## 1. Introduction

The use of garlic (*Allium sativum* L.) as a spice in the food industry is common [1,2]. Most of the characteristic volatiles in garlic contain sulfur [3]. In raw garlic, the primary sulfur-containing compound is alliin. When the garlic is crushed, chopped or otherwise disrupted, the enzyme alliinase converts alliin into allicin, which is responsible for the pungent aroma of fresh garlic. However, allicin is unstable and breaks down quickly to form allyl methyl disulfide and diallyl disulfide. Diallyl disulfide is reduced to form allyl mercaptan. Allyl mercaptan is methylated to form allyl methyl sulfide [4,5]. These volatiles are responsible for the malodorous nature of garlic [4,5]. Diallyl disulfide and allyl mercaptan are the volatiles with the highest concentration in garlic [5].

Various foods, such as vegetables, fruits and herbs, and beverages have been used to deodorize garlic volatiles. Treatments including mint leaves (spearmint and peppermint) [6,7], pear, loquat, peach, plum, prune, apricot, cherry, grape, chicory, udo, perilla, burdock, potato, eggplant [7], dried and fresh herbs [8], lemon juice, green tea [9], spinach, parsley [9,10], basil [7,10], kiwi [10], raw egg, boiled rice [10], mushrooms [7,10], apple [7,9,11], raw lettuce [7,10,11], milk [10,12] and oil [12] have been used to reduce the concentration of volatiles in garlic.

Several different mechanisms have been proposed for deodorization. The raw vegetables, fruits and herbs were proposed to cause deodorization due to their enzymatic and phenolic activity [6,8,9,11]. The enzyme polyphenol oxidase (PPO) catalyzes the oxidation of the sulfur compounds responsible for the odor of the garlic [6,8,9,11]. Milk deodorizes due to the water and fat present in it [10,12]. The milk fat deodorizes the garlic volatiles by creating a hydrophobic environment [12]. The water reduces garlic volatiles by partitioning volatiles between phases [12]. Proteins deodorize by the adsorption of volatiles through hydrogen and hydrophobic bonding [13,14]. The protein casein is proposed to interact through disulfide interactions or sulfhydryl bonds [12,13]. Whey protein interacts through hydrophobic interactions [13].

Yogurt is a fermented milk product that is popular among consumers for its health-promoting, nutritional and sensorial properties. Yogurt has 0.15–5% fat, 3–10% protein and 70–87% water depending on the type of yogurt [15]. However, yogurt has not been tested for its ability to reduce the concentration of the sulfur volatiles in garlic. Therefore, the objective of this study is to test yogurt and to understand the mechanism behind the deodorization of both raw and fried garlic with yogurt and its components.

## 2. Results and Discussion

### 2.1. Effect of Frying on Raw Garlic Volatiles

The volatiles responsible for most of the pungent odor in both raw and fried garlic are allyl methyl disulfide, diallyl disulfide, allyl mercaptan and allyl methyl sulfide [12,16]. Allyl methyl disulfide has a garlic- like, savory odor [17]. Diallyl disulfide has a pungent, garlic- like odor with an odor threshold of 0.22 ppb [17]. Allyl mercaptan has a sulfurous odor with an odor threshold of 0.05 ppb [18]. Allyl methyl sulfide has a garlic- like, savory odor with an odor threshold of 0.14 ppb [17]. All of these volatiles were above their threshold level in both raw and fried garlic, which means the pungent odor of these volatiles is perceived or detected by the human nose. Frying garlic resulted in a significant reduction in almost all sulfur volatile compounds (Table 1). The levels of allyl methyl disulfide, diallyl disulfide, allyl mercaptan and allyl methyl sulfide were decreased by 75 to 99% by frying (Table 1). Other studies have also shown that frying garlic causes a significant loss of sulfur volatile compounds including diallyl disulfide, allyl methyl sulfide and allyl mercaptan [19,20,21]. Frying breaks down the sulfur-containing compounds in garlic, leading to a decrease in its characteristic malodorous odor-producing volatiles.

However, a few volatiles increase when garlic is fried. 2-vinyl-4H-1,3-dithiin increased in fried garlic (Table 1). Other studies have also shown an increase in 2-vinyl-4H-1,3-dithiin [19,20,21]. Allicin transforms into vinyl dithiins in non-polar solvent so 2-vinyl-4H-1,3-dithiin was expected to increase when the garlic was fried in oil [22]. 2,3-Dimethylpyrazine and pyridine also significantly increased in fried garlic (Table 1). When garlic is fried, nitrogen-containing compounds (pyridine and pyrazine) are formed by decomposing sulfur-containing compounds during heating due to the Maillard reaction [19,23,24].

### 2.2. Effect of Yogurt on Raw and Fried Garlic Volatiles

Because of variability in raw material and the frying process, each treatment was compared to a control made from the same batch of garlic to accurately determine the extent of deodorization by the treatment. The addition of yogurt significantly reduced the concentration of almost all of the volatiles in both raw and fried garlic (Table 2). Yogurt produced a higher percent reduction in raw garlic than fried garlic. In raw garlic, the concentration of diallyl disulfide, allyl mercaptan, allyl methyl sulfide and allyl methyl disulfide were reduced by 99%, while in fried garlic, they were reduced by 82–94%. The volatiles were at a much higher concentration in the raw garlic than in the fried, which may explain why a greater reduction was achieved in the raw garlic.

The yogurt interacts with the sulfur volatiles in garlic, decreasing the concentration of garlic volatiles. We theorize that the reduction was due to the fat, proteins and water present in the yogurt. When garlic is treated with milk, there is a reduction in the concentration of sulfur-containing volatiles [12,13]. The authors proposed that fat and water were responsible for deodorizing the garlic volatiles [12,13,14]. As expected, yogurt shows a similar effect on garlic volatiles to milk. Therefore, this study further focused on each of the components of the yogurt separately to understand the impact of each component on garlic volatiles.

### 2.3. Effect of Fat Versus Protein on Raw and Fried Garlic Volatiles 

Next, 9% butter fat or milk protein were added to raw garlic and fried garlic (Figure 1 and Figure 2). The concentrations of almost all sulfur- containing volatiles were greatly reduced; thus, both fat and protein are excellent treatments for reducing the malodorous odor produced by garlic. 

Whether fat or protein was more effective at deodorization depended on the volatile (Figure 1 and Figure 2). For diallyl disulfide and allyl methyl disulfide, butter reduced the concentration more than milk protein in both raw and fried garlic (Figure 1 and Figure 2). Diallyl disulfide (LogP = 2.2) and allyl methyl disulfide (LogP = 1.6) are more hydrophobic volatiles so should have stronger interactions with fat than protein [12]. There was no significant difference for allyl mercaptan (LogP = 0.5) and allyl methyl sulfide (LogP = 1.5), but protein appeared to reduce the concentration more than fat in both raw and fried garlic. Allyl mercaptan and allyl methyl sulfide are less hydrophobic in nature so they should have stronger interactions with the protein than with the fat [12].

### 2.4. Effect of Water and Protein on Raw and Fried Garlic Volatiles

Water (pH 7) and water adjusted to pH 4.4 were added to garlic to isolate the effect of water and pH (Table A1 and Table A2, Figure 3 and Figure 4). When water was added to the garlic, volatile levels decreased, as expected (Table A1 and Table A2). The reduction in the concentration of volatiles is due to the partitioning of volatiles between phases [12]. The headspace concentrations of these volatiles were lower when water was added to the garlic because the volatiles were partitioning between the garlic, water and gas phases, whereas the volatiles in garlic without water only partitioned between the garlic and gas phases [12,25]. Changing the pH of the water from 7 to 4.4 created no significant difference in the deodorization of the volatiles; thus, the pH, in and of itself, did not affect the volatiles.

Yogurt (pH 4.4) deodorized the garlic, resulting in a high level of reduction (99%) in allyl mercaptan, allyl methyl disulfide, allyl methyl sulfide and diallyl disulfide (Table A1 and Table A2). However, increasing the pH of the yogurt to pH 7 decreased the deodorization (Figure 3 and Figure 4). Figure 3 and Figure 4 show that changing the pH of water between 4.4 and 7 had no effect; thus, this effect was not due to the pH itself. Instead, the pH changed the conformation of the protein, which changed the protein’s ability to bind the volatiles. The functional properties of proteins are highly dependent on their three-dimensional structure, which is affected by pH [13,26,27]. At casein’s isoelectric point, pH 4.6, casein forms a three-dimensional protein network called micellar casein [28]. Micellar casein binds to denatured whey protein via disulfide bridges [28,29]. We propose that the mechanism responsible for deodorization is the casein micelle–whey protein complex binding to the volatiles. The hydrophobic and reactive thiol groups of denatured whey protein (i.e., α-lactalbumin and β-lactoglobulin) are exposed, which allows them to bind with the volatiles [29,30,31,32]. Casein also binds to the volatiles through hydrophobic bonding, hydrogen bonding, covalent interaction, redox reactions or interchange reactions between the thiol and disulfide groups of the sulfur volatiles and the sulfhydryl groups of the protein [13,33,34,35,36]. This casein micelle–whey protein complex binds to more volatiles than either protein separately.

As the pH is increased to 7, the casein micelle–whey protein complex breaks down. At pH 7, casein carries a net negative charge, which results in an electrostatic repulsion between the casein molecules, preventing the formation of the micelle structure. Whey protein is in a stable soluble state, which restricts its binding ability [27,37]. Due to these factors, both proteins were less able to bind the volatiles at pH 7.

The yogurt (pH 4.4) was also heated to 80 °C to denature the proteins. Heating the yogurt resulted in less deodorization of garlic volatiles as compared to unheated yogurt (pH 4.4). When yogurt is heated, the increase in kinetic energy disrupts the bonds in the proteins, resulting in the unfolding and denaturation of both casein and whey protein [13,38]. This decreases the ability of the proteins to bind to volatiles, thus increasing the concentration of volatiles in the headspace.

### 2.5. Effect of Different Proteins on Raw and Fried Garlic Volatiles

Two forms of whey protein, casein or milk protein, were mixed with either raw or fried garlic. All of the proteins caused significant deodorization of almost all volatiles (Table A3 and Table A4, Figure 5 and Figure 6). Proteins react with volatiles through a number of mechanisms, including reversible binding, such as hydrogen bonds, hydrophobic interactions and ionic bonds, and irreversible binding via covalent linkages [13].

The type of protein that was most effective at deodorization was different for raw versus fried garlic (Figure 5 and Figure 6). In raw garlic, whey protein created the greatest reduction in volatile concentration, followed by casein then milk protein. In fried garlic, whey protein was less effective than in raw garlic. Frying causes both the Maillard reaction and oxidation of fatty acids in the frying oil, producing many new compounds such as aldehydes, ketones and alcohols. We propose that the whey protein may have preferentially bound to the compounds formed during frying over the sulfur volatiles. The whey protein caused deodorization in both samples, but it was more effective in the raw garlic. 

The form of the protein generally made no significant difference, but when there was a difference, the concentrate was more effective than the isolate. At first, it would logically seem that the isolate, with a higher protein content, would create greater deodorization. However, during the production of concentrates, the proteins undergo less processing than for isolates. Additional processes for the purification and separation of isolates are used. Due to this additional processing, some of the protein structure is changed or denatured, and thus the structures of the proteins are more intact in concentrates than in isolates [39]. The more intact conformation of the proteins in concentrates may have made them more effective in binding the volatiles than the isolates. 

### 2.6. Effect of Quantity of Butter Fat on Raw and Fried Garlic Volatiles 

The quantity of butter also affects the deodorization of malodor- producing garlic volatiles. Garlic was mixed with a solution of water containing 3, 10, 20, 40 or 80% butter. Both water and fat reduce volatile levels. The addition of butter to the water significantly increased the deodorization of both raw and fried garlic sulfur volatiles (Table A5 and Table A6, Figure 7 and Figure 8). In both raw and fried garlic, the reduction in the concentration of these four volatiles significantly increased as the butter content increased to 80% (Figure 7 and Figure 8). In raw garlic, many of the treatments were not significantly different. However, in fried garlic, increasing the butter from 3 to 10% created a large decrease, with a significant additional decrease as the butter content further increased. Increasing butter content decreases the concentration of the volatiles by creating a more hydrophobic environment. Sulfur volatiles interact with fat through hydrophobic interactions involving the nonpolar regions of the sulfur volatiles and the fat molecules [9,12,14,40,41]. When butter is added to garlic, these volatiles prefer to reside in the fat phase, thereby lowering their concentration in the headspace [22]. Therefore, using fat is an effective way to minimize the pungent odor of garlic.

## 3. Materials and Methods

### 3.1. Garlic Sample Preparation

Garlic cloves (Walmart, Columbus, OH, USA) were peeled by hand. Using a knife, peeled garlic cloves were cut into slices of approximately 1 mm thickness. Sliced garlic was held for 30 min at room temperature before frying or addition of treatments. Fried garlic was stir-fried in hot canola oil (Kroger, Columbus, OH, USA), (125 °C). A temperature drop was seen immediately after the addition of chopped garlic into the hot oil. The frying continued until the temperature reached 125 °C again, at approximately 12 min. The oil was removed from the fried garlic using tissues and the garlic was cooled at room temperature for 15 min.

### 3.2. Treatment Preparation

Different treatments were used for the determination of the deodorization of the garlic. Because of variability in raw material and the frying process, one batch of garlic was prepared and used for one set of treatments with its control. Thus, there was a different control for each set of treatments (fat and protein, pH of water and yogurt, types of protein, amount of fat). For water, 100 mL water at pH 4.4 or pH 7 and 100 g whole milk plain yogurt (Dannon, Target, Columbus, OH, USA) at pH 4 or pH 7 were used. To adjust the pH, hydrochloric acid was used for acidic pH and sodium hydroxide was used for basic pH. Heating involved briefly immersing the raw or fried garlic mixed with whole fat yogurt (Dannon, Target, Columbus, OH, USA) packed in a Ziplock bag in boiling water, followed by rapid cooling. For protein, a mixture of 50 mL water and 50 g non-fat dry milk powder (Kroger, Columbus, OH, USA) was used. For different proteins, 9% whey protein isolate (Mill Haven Foods, New Lisbon, WI, USA), whey protein concentrate (Mill Haven Foods, New Lisbon, WI, USA), calcium caseinate (Mill Haven Foods, New Lisbon, WI, USA), micellar casein (Mill Haven Foods, New Lisbon, WI, USA), milk protein isolate (Mill Haven Foods, New Lisbon, WI, USA) or milk protein concentrate (Mill Haven Foods, New Lisbon, WI, USA) was used. The proteins were rehydrated by mixing using a magnetic stirrer for 4 h. For the quantity of fat, a 100 mL solution of water containing 3, 10, 20, 40 or 80% of unsalted butter (Kroger, Columbus, OH, USA) was heated, mixed, then cooled to room temperature before using. The raw or fried garlic (6 g) mixed with 100 mL of each treatment was placed in a 500 mL Pyrex bottle. The raw or fried garlic (6 g), was used as a control.

### 3.3. Selected-Ion Flow-Tube Mass Spectrometry (SIFT-MS) Headspace Analysis

The raw or fried garlic (6 g) was mixed with 100 mL of each treatment. The sample was placed in a 500 mL Pyrex bottle capped with a polytetrafluoroethylene (PTFE)- faced silicone septa cap and held for 30 min at room temperature. The headspace volatiles of the garlic sample were measured via an 18-gauge passivated needle connected to the inlet of the selected-ion flow-tube mass spectrometry (SIFT-MS) machine (SYFT Voice200ultra, Syft Ltd., Christchurch, New Zealand) using a selected ion mode (SIM) scan at 20 °C ± 2 and the SIFT-MS headspace method. The SIFT-MS was calibrated with the standard gases benzene, toluene, isobutane, ethylene, tetrafluorobenzene, hexafluorobenzene and octafluorobenzene. Syft VOICE-200 software (v.1.4.9.17754, Syft Technologies Ltd., Christchurch, New Zealand) was used to analyze the volatiles. 2-vinyl-4H-1,3-dithiin is a mixture of 2-vinyl-4H-1,3-dithiin and 3-vinyl-4H-1,2-dithiin, and 2,3-dimethylpyrazine is a mixture of 2,3-dimethylpyrazine, 2,5-dimethylpyrazine and 2,6-dimethylpyrazine. Three replicates were conducted for each deodorizing agent. A blank was used before testing the samples. An empty 500 mL Pyrex bottle was used as the blank. The scanning time for each headspace reading was 95 s. Volatiles tested are given in Table 3.

### 3.4. Statistical Analysis

The data were coded, entered and analyzed using JMP^®^ Pro version 16.0.0 (512340) (Statistical Discovery, Cary, NC, USA). Three replicates of data for each test were obtained. Data were analyzed using one-way analysis of variance (ANOVA) and comparison of all pairs was performed using Fisher’s least significance difference (LSD). Significance was defined as *p* ≤ 0.05.

## 4. Conclusions

Frying garlic significantly reduced the concentration of almost all sulfur volatile compounds. Yogurt also significantly reduced the headspace concentration of all major odor- producing garlic volatiles, including allyl methyl disulfide, diallyl disulfide, allyl mercaptan and allyl methyl sulfide, in both raw and fried garlic. The components responsible for the deodorization of the volatiles were fat, protein and water. At the same concentration, fat was a more effective treatment for the reduction in diallyl disulfide and allyl methyl disulfide, whereas protein seemed to be a more effective treatment for the reduction in allyl methyl sulfide and allyl mercaptan. The proteins whey protein isolate, whey protein concentrate, micellar casein, calcium casein, milk protein isolate and milk protein concentrate were all effective in deodorizing the garlic volatiles. The concentrate form of proteins was more effective than the isolate form in the deodorization of volatiles. pH has no effect on the deodorization of garlic volatiles but affects the proteins by changing the conformation of the protein, which changes the binding of volatiles. Heating yogurt decreases its ability to deodorize the volatiles as heating denatures proteins, which results in weakening of the bonds of the proteins, leading to less deodorization. The amount of fat also affects the concentration of volatiles, with higher quantities of fat making a more hydrophobic environment, leading to more effective deodorization. The results suggest that consuming yogurt or foods with proteins and fat may help reduce the bad odor produced by garlic. Follow- up studies will focus on the effectiveness of these treatments at deodorizing the breath after the consumption of garlic. Protein and fat can be used as a natural deodorizer to mask the strong smell of garlic in processed foods. New products can be formulated to minimize the pungent odor of garlic using fat and dairy proteins.

## Figures and Tables

**Figure 1 molecules-28-05714-f001:**
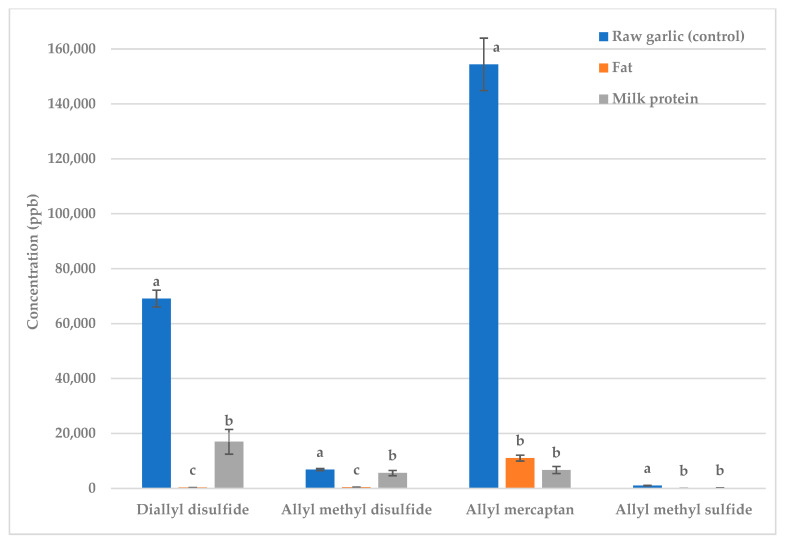
Raw garlic: effect of fat vs. protein on key sulfur volatiles. Treatments within the same volatile with different letters are significantly different (*p* < 0.05).

**Figure 2 molecules-28-05714-f002:**
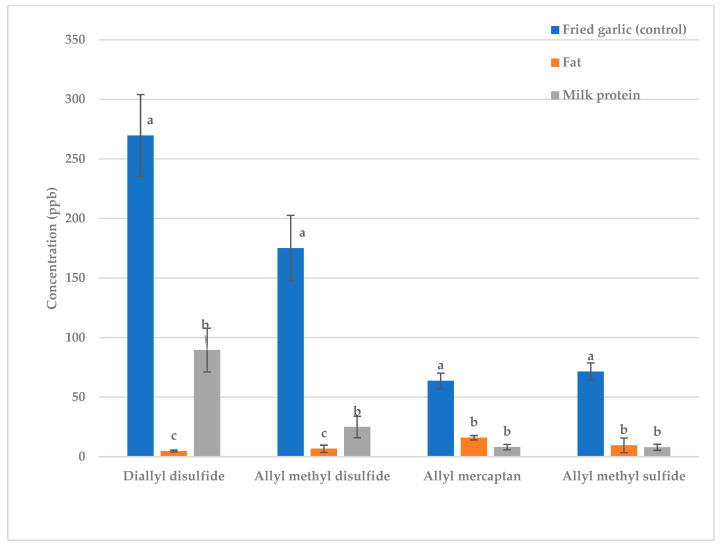
Fried garlic: effect of fat vs. protein on key sulfur volatiles. Treatments within the same volatile with different letters are significantly different (*p* < 0.05).

**Figure 3 molecules-28-05714-f003:**
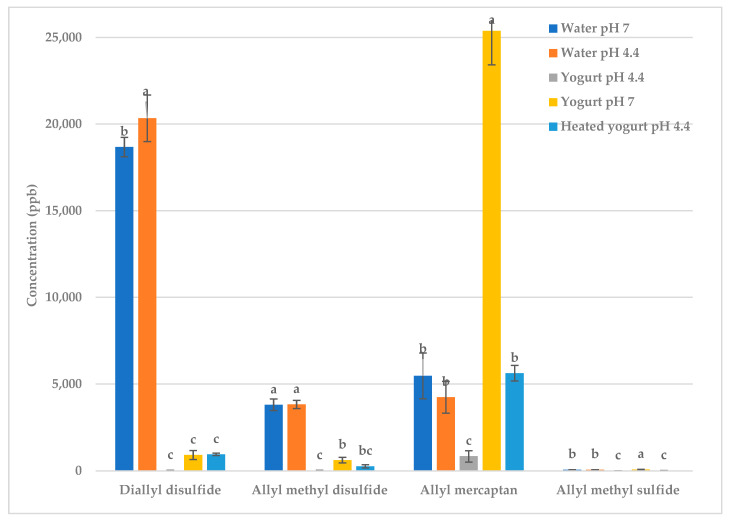
Raw garlic: effect of water, pH and heating on key sulfur volatiles. Treatments within the same volatile with different letters are significantly different (*p* < 0.05).

**Figure 4 molecules-28-05714-f004:**
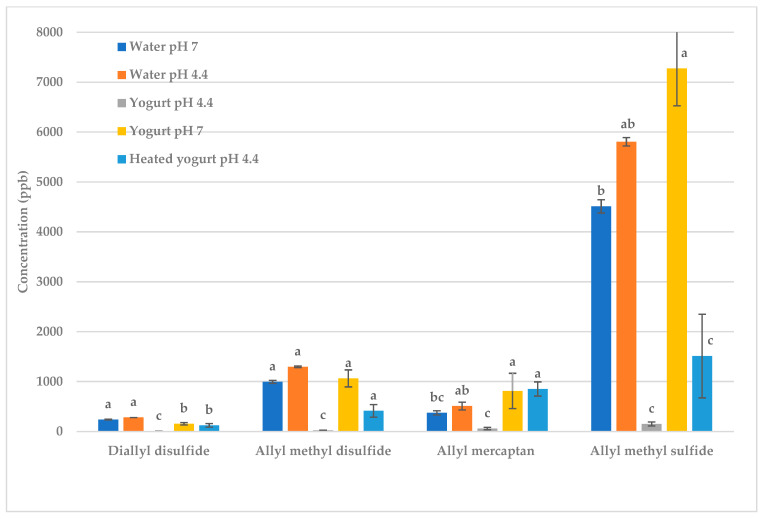
Fried garlic: effect of water, pH and heating on key sulfur volatiles. Treatments within the same volatile with different letters are significantly different (*p* < 0.05).

**Figure 5 molecules-28-05714-f005:**
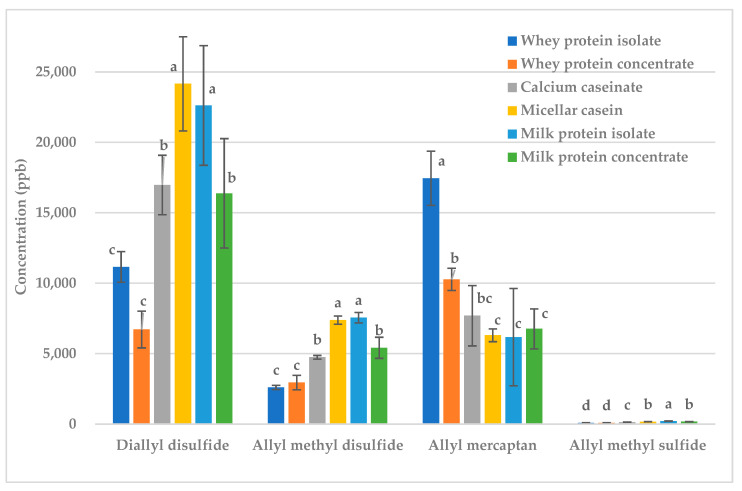
Raw garlic: effect of different proteins on key sulfur volatiles. Treatments within the same volatile with different letters are significantly different (*p* < 0.05).

**Figure 6 molecules-28-05714-f006:**
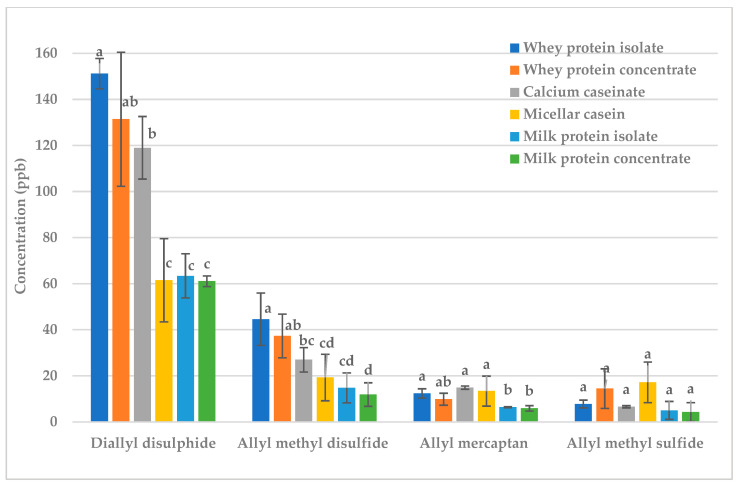
Fried garlic: effect of different proteins on key sulfur volatiles. Treatments within the same volatile with different letters are significantly different (*p* < 0.05).

**Figure 7 molecules-28-05714-f007:**
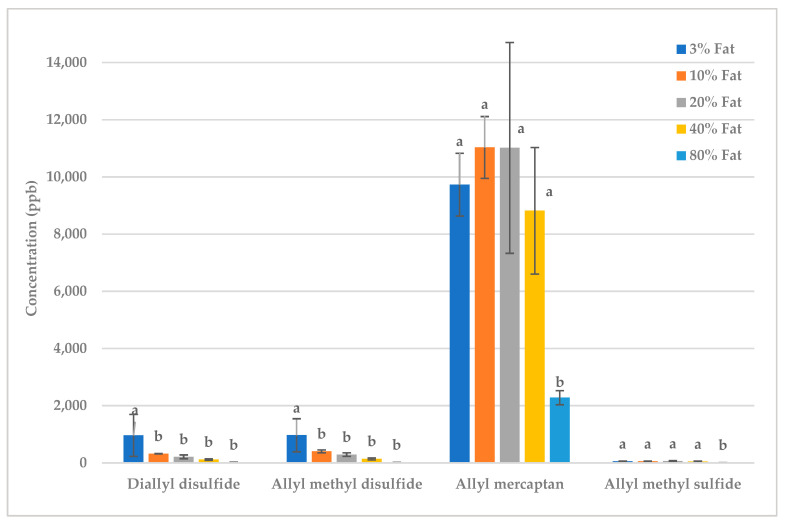
Raw garlic: effect of quantity of fat on key sulfur volatiles. Treatments within the same volatile with different letters are significantly different (*p* < 0.05).

**Figure 8 molecules-28-05714-f008:**
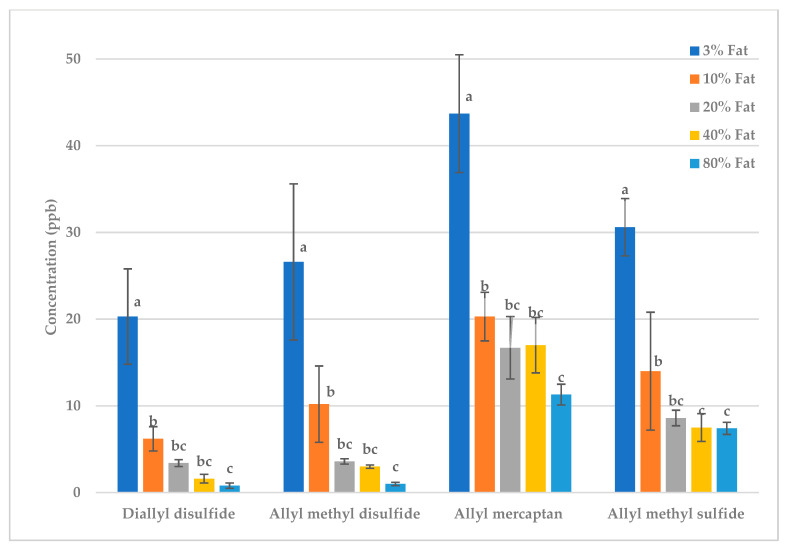
Fried garlic: effect of quantity of fat on key sulfur volatiles. Treatments within the same volatile with different letters are significantly different (*p* < 0.05).

**Table 1 molecules-28-05714-t001:** Effect of frying on garlic volatiles (concentration (ppb) ± SD).

Volatiles	Raw Garlic	Fried Garlic
1,3-dithiane	1240 ± 879	65.9 ± 31.8
2-ethylpyrazine	7.20 ± 5.83	11.0 ± 4.5
2-methyl-2-butenal	60.9 ± 49.1	23.6 ± 8.67
2-methylbenzaldehyde	5.97 ± 3.37	1.36 ± 0.77
2-vinyl-4H-1,3-dithiin	4.21 ± 2.94	8.26 ± 3.26
2,3-dimethylpyrazine	4.81 ± 3.89	15.5 ± 10.7
2,5-dimethylthiophene	134 ± 100	9.85 ± 1.62
Acetaldehyde	19,100 ± 11,300	7990 ± 3390
Allicin	1330 ± 1190	81.4 ± 42.1
Allyl mercaptan	116,000 ± 20,200	121 ± 8
Allyl methyl disulfide	13,400 ± 8780	79.8 ± 29.2
Allyl methyl sulfide	1380 ± 385	73.3 ± 9.3
Allyl methyl tetrasulfide	2.56 ± 1.25	1.66 ± 0.57
Allyl methyl thiosulfinate	89.4 ± 73.8	8.06 ± 2.02
Allyl methyl trisulfide	35.7 ± 18.5	7.46 ± 0.76
Aniline	2.56 ± 1.56	5.93 ± 3.34
Diallyl disulfide	26,200 ± 19,000	69.1 ± 43.5
Diallyl sulfide	1900 ± 711	51.8 ± 5.7
Diallyl tetrasulfide	2.80 ± 2.12	1.18 ± 0.51
Diallyl trisulfide	236 ± 175	14.7 ± 4.2
Dimethyl disulfide	554 ± 367	47.9 ± 24.8
Dimethyl sulfide	981 ± 692	384 ± 159
Dimethyl thioether	1130 ± 794	447 ± 174
Dimethyl thiosulfinate	10.7 ± 4.7	3.63 ± 0.08
Dimethyl trisulfide	10.9 ± 4.94	9.58 ± 4.06
Dipropyl sulfide	5130 ± 4240	16.3 ± 4.1
Methyl mercaptan	24,400 ± 3580	55.9 ± 7.15
Methyl propyl disulfide	1970 ± 1350	19.5 ± 9.9
Pyridine	2.42 ± 1.65	6.91 ± 3.19
Trimethylpyrazine	493 ± 344	6.14 ± 2.86

**Table 2 molecules-28-05714-t002:** Effect of yogurt on raw and fried garlic (concentration (ppb) ± SD).

Volatiles	Raw Garlic	Yogurt + Raw Garlic	Fried Garlic	Yogurt + Fried Garlic
1,3-dithiane	4020 ± 1770	20.4 ± 6.6	203 ± 2	16.6 ± 15.6
2-ethylpyrazine	29.9 ± 15.7	0.67 ± 0.10	25.4 ± 4.6	1.08 ± 0.76
2-methyl-2-butenal	93.0 ± 37.9	2.36 ± 0.39	36.3 ± 1.5	3.59 ± 2.81
2-methylbenzaldehyde	51.6 ± 25.1	4.20 ± 0.50	2.32 ± 0.09	2.91 ± 2.11
2-vinyl-4H-1,3-dithiin	8.59 ± 3.22	3.18 ± 1.02	13.0 ± 0.2	1.76 ± 0.97
2,3-dimethylpyrazine	38.8 ± 17.1	0.44 ± 0.06	51.9 ± 3.1	0.95 ± 0.84
2,5-dimethylthiophene	401 ± 212	2.59 ± 0.54	19.9 ± 0.7	1.77 ± 1.10
Acetaldehyde	10,400 ± 617	9210 ± 455	3380 ± 322	6770 ± 5310
Allicin	93.9 ± 39.9	3.66 ± 2.73	3.86 ± 1.03	0.41 ± 0.24
Allyl mercaptan	110,000 ± 28,600	832 ± 324	169 ± 5	29.0 ± 27.8
Allyl methyl disulfide	20,100 ± 6150	23.2 ± 8.1	350 ± 18	18.1 ± 16.2
Allyl methyl sulfide	965 ± 275	8.39 ± 0.34	742 ± 20	75.3 ± 90.1
Allyl methyl tetrasulfide	9.34 ± 4.43	2.12 ± 0.30	2.90 ± 0.16	1.27 ± 0.17
Allyl methyl thiosulfinate	50.8 ± 23.8	0.98 ± 0.69	28.9 ± 1.9	1.29 ± 0.83
Allyl methyl trisulfide	35.3 ± 14.6	1.13 ± 0.15	54.0 ± 2.4	3.54 ± 2.78
Aniline	9.38 ± 3.67	0.49 ± 0.09	15.7 ± 2.6	0.29 ± 0.27
Diallyl disulfide	102,000 ± 56,900	28.3 ± 8.7	84.9 ± 2.3	4.64 ± 3.92
Diallyl sulfide	1720 ± 841	5.96 ± 2.40	250 ± 14	18.1 ± 21.8
Diallyl tetrasulfide	13.3 ± 6.03	3.38 ± 0.11	1.44 ± 0.23	1.20 ± 0.66
Diallyl trisulfide	981 ± 522	2.35 ± 0.79	28.4 ± 1.8	2.11 ± 1.49
Dimethyl disulfide	487 ± 71.3	2.93 ± 0.92	293 ± 3	23.1 ± 27.6
Dimethyl sulfide	2870 ± 1350	69.7 ± 4.21	151 ± 16	37.0 ± 33.8
Dimethyl thioether	2990 ± 950	77.6 ± 7.43	182 ± 43	43.4 ± 40.3
Dimethyl thiosulfinate	13.5 ± 6.8	1.00 ± 0.51	15.8 ± 0.2	0.81 ± 0.26
Dimethyl trisulfide	10.6 ± 1.6	2.24 ± 0.53	110 ± 15	7.00 ± 5.40
Dipropyl sulfide	7900 ± 2530	30.5 ± 4.40	194 ± 15	23.6 ± 19.2
Methyl mercaptan	16,600 ± 1300	449 ± 172	218 ± 29	102 ± 103
Methyl propyl disulfide	3410 ± 1130	4.92 ± 0.83	71.1 ± 1.7	4.07 ± 3.59
Pyridine	20.8 ± 6.6	0.78 ± 0.11	16.6 ± 0.8	0.76 ± 0.65
Trimethylpyrazine	958 ± 364	1.80 ± 0.33	24.0 ± 1.5	1.38 ± 0.96

**Table 3 molecules-28-05714-t003:** Properties of volatiles tested in selected-ion flow-tube mass spectrometry (SIFT-MS) headspace analysis.

Volatile Compound	Ion Product	Reagent Ion	*m*/*z*	Reaction Rate (k) 10^−9^ cm^3^ s^−1^
1-3-dithiane	C_4_H_7_S^+^	O_2_^+^	87	2.3
2-ethylpyrazine	C_6_H_8_N_2_.H^+^	H_3_O^+^	109	3
2-Methyl-2-butenal	C_5_H_7_O^+^	NO^+^	83	4
2-methylbenzyaldehyde	C_8_H_7_O^+^	NO^+^	119	3.3
2-vinyl-4H-1,3-dithiin	C_6_H_8_S_2_^+^	NO^+^	144	2.4
2,3-dimethylpyrazine	C_6_N_2_H_8_.H^+^	H_3_O^+^	109	3.4
2,5-dimethylthiophene	C_6_H_8_S.H^+^	H_3_O^+^	113	3
Acetaldehyde	CH_3_CO^+^	NO^+^	43	6
	C_2_H_5_O^+^	H_3_O^+^	45	3.7
Allicin	C_6_H_10_OS_2_	NO^+^	162	2.4
Allyl mercaptan	C_3_H_6_S	NO^+^	74	2.4
	C_3_H_6_S.H^+^	H_3_O^+^	75	2.6
Allyl methyl disulfide	C_4_H_8_S_2_	NO^+^	120	2.4
	C_4_H_8_S_2_.H^+^	H_3_O^+^	121	2.6
Allyl methyl sulfide	C_4_H_8_S^+^	NO^+^	88	2.5
	C_4_H_8_S.H^+^	H_3_O^+^	89	3
Allyl methyl tetrasulfide	C_4_H_8_S_4_^+^	NO^+^	184	2.4
Allyl methyl thiosulfinate	C_4_H_8_S_2_O	NO^+^	136	2.4
Allyl methyl trisulfide	C_4_H_8_S_3_^+^	NO^+^	152	2.4
Aniline	C_6_H_5_NH_2_.H^+^	H_3_O^+^	94	2.8
Diallyl disulfide	(C_3_H_5_)_2_S_2_.H^+^	H_3_O^+^	147	3
Diallyl sulfide	(C_3_H_5_)_2_S.H^+^	H_3_O^+^	115	2.9
Diallyl tetrasulfide	C_6_H_10_S_4_^+^	NO^+^	210	2.4
Diallyl trisulfide	C_6_H_10_S_3_	NO^+^	178	2.4
Dimethyl disulfide	(CH_3_)_2_S_2_.H_+_	H_3_O^+^	95	2.6
Dimethyl sulfide	(CH_3_)_2_S^+^	O_2_^+^	62	2.2
Dimethyl thioether	(CH_3_)_2_S^+^	O_2_^+^	62	2.2
	(CH_3_)_2_S^+^	NO^+^	62	2.2
Dimethyl thiosulfinate	C_2_H_6_OS_2_^+^	NO^+^	110	2.4
Dimethyl trisulfide	C_2_H_6_S_3_^+^	NO^+^	126	1.9
Dipropyl sulfide	C_3_H_4_S^+^	O_2_^+^	72	2.4
Methyl mercaptan	CH_4_S.H^+^	H_3_O^+^	49	1.8
Methyl propyl disulfide	C_4_H_10_S_2_^+^	NO^+^	122	1.5
	C_4_H_10_S_2_.H^+^	H_3_O^+^	123	2.5
Pyridine	C_5_H_5_N.H^+^	H_3_O^+^	80	3.3
Trimethylpyrazine	C_7_H_10_N_2_^+^	O_2_^+^	122	2.5

## Data Availability

The data presented in this study are available in Appendix A.

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
