# Peer review of "Effect of Yogurt and Its Components on the Deodorization of Raw and Fried Garlic Volatiles"

_molecules, 2023, doi:10.3390/molecules28155714_

Round 1
Reviewer 1 Report
The manuscript “Effect of yoghurt and its components on the deodorization of raw and fried garlic volatiles” by Kaur and Barringer describes the ability of yoghurt in deodorization of raw and fried garlic by measuring the concentrations of volatile compounds in differently treated raw and fried garlic. The authors do not offer novel explanations for their results. For example, the different effects of whey and casein on raw and fried garlic deodorization are not explained even as a hypothesis. Therefore, their study's novelty is unclear since they do not provide novel explanations. Instead, they only present measured concentrations and cite literature data for the effects of fat, water and proteins. Inconsistencies in the concentrations of volatile compounds between different figures and tables are present throughout the manuscript.
Some specific comments:
Some inconsistencies in the value were observed in Tables 1 and 2. Concentrations of allicin in the two tables differ drastically. Namely, Table 1 shows an allicin concentration of 1330 ppb in raw garlic and 81.4 in fried garlic, while Table 2 shows a concentration of 93.9 ppb in raw garlic and 3.86 in fried garlic.
The concentration of diallyl disulfide in raw garlic in Table 2 is not written correctly. Shouldn’t it be either 102,000 ± 56,900 or 10,200 ± 5,690? Similar goes for allyl mercaptan concentrations in the same table. Please check. If the reviewer is wrong and the concentrations are indeed as written, with standard deviations higher than the mean value, the standard deviation may not be the best choice for expression of the standard of data set variation and the median might be more adequate.
Concentrations of diallyl disulphide and allyl mercaptan in raw garlic are inconsistent in Table 1, Table 2 and Figure 1.
The concentration of allicin in the control sample is lower than in the sample to which heated yogurt was added (Table A1). How do authors explain that?
Concentrations of control samples (no treatment) should be presented in Figures 3 and 4 to allow better comparison. Also, standard deviations are not fully visible in some samples in the same figures, but also in Figure 5, Figure 7 and Figure 8. Please revise.
Control samples also need to be included in Figures 5-8.
English language is fine.
Author Response
The authors do not offer novel explanations for their results. For example, the different effects of whey and casein on raw and fried garlic deodorization are not explained even as a hypothesis. Therefore, their study's novelty is unclear since they do not provide novel explanations. Instead, they only present measured concentrations and cite literature data for the effects of fat, water and proteins.
We have added our hypothesis in section 2.5. We also offer our hypothesis in section 2.3 and 2.4.
Inconsistencies in the concentrations of volatile compounds between different figures and tables are present throughout the manuscript.
All the tests have different controls so they have different concentrations. This information has been added to the methods section 3.2 and the results 2.2
Some specific comments:
Some inconsistencies in the value were observed in Tables 1 and 2. Concentrations of allicin in the two tables differ drastically. Namely, Table 1 shows an allicin concentration of 1330 ppb in raw garlic and 81.4 in fried garlic, while Table 2 shows a concentration of 93.9 ppb in raw garlic and 3.86 in fried garlic.
Each table has a different control so the tables have different values. This information has been added to the methods section 3.2 and the results 2.2
The concentration of diallyl disulfide in raw garlic in Table 2 is not written correctly. Shouldn’t it be either 102,000 ± 56,900 or 10,200 ± 5,690? Similar goes for allyl mercaptan concentrations in the same table. Please check. If the reviewer is wrong and the concentrations are indeed as written, with standard deviations higher than the mean value, the standard deviation may not be the best choice for expression of the standard of data set variation and the median might be more adequate.
The values were not written correctly. The correction has been made.
Concentrations of diallyl disulphide and allyl mercaptan in raw garlic are inconsistent in Table 1, Table 2 and Figure 1.
All the tests have different controls so they have different concentrations. This information has been added to the methods section 3.2 and the results 2.2
The concentration of allicin in the control sample is lower than in the sample to which heated yogurt was added (Table A1). How do authors explain that?
We made a copy paste error. We re-checked all of the values in the tables. It is corrected in the table now.
Concentrations of control samples (no treatment) should be presented in Figures 3 and 4 to allow better comparison.
The control was removed to show the difference between the treatments. The values of the control sample is very high as compared to the treatments, which means you cannot see the difference between the treatments when the control is present. The tables in the appendix have been added for comparison of treatments with the control.
Also, standard deviations are not fully visible in some samples in the same figures, but also in Figure 5, Figure 7 and Figure 8. Please revise.
All the figures have been revised
Control samples also need to be included in Figures 5-8.
The control was removed to show the difference between the treatments. The values of the control sample is very high as compared to the treatments, which means you cannot see the difference between the treatments when the control is present. The tables in the appendix have been added for comparison of treatments with the control.
Reviewer 2 Report
MoleculesArticle
Effect of yoghurt and its components on the deodorization of raw and fried garlic volatiles
Authors have presented on the Effect of yoghurt and its components on the deodorization of raw and fried garlic (Allium sativum L.) volatiles.
Garlic contains sulfur volatiles that cause bad odor after consumption, while yoghurt deodorizes these volatiles and
therefore, this has become the objective of this study to understand how the components of yoghurt cause deodorization.
Authors have thoroughly tested yogurt understanding the mechanism behind the deodorization of both raw and fried garlic with yogurt and its components
The manuscript is very informative and with good Food and Nutrition applications.
Yoghurt or yogurt?
Frying garlic significantly reduced the concentration of almost all sulfur volatile compounds.
Yoghurt also significantly reduced the headspace concentration of all major odor-producing garlic volatiles
including allyl methyl disulfide, diallyl disulfide, allyl mercaptan, and allyl methyl sulfide, in both raw and fried garlic.
The components responsible for the deodorization of the volatiles were fat, protein and water.
The title was good
Well written and structured.
Contain adequate tables and several well-drawn figures, equally informative
Literature may be increased slightly
Conclusions are evidenced by data
Elaborate on Results and discussion, may be clear mechanism if any
References can be increased are just adequate, authors may revisit it.
Recheck for English grammar.
Accept the manuscript for publication with minor mandatory changes
With Regards,
MoleculesArticle
Effect of yoghurt and its components on the deodorization of raw and fried garlic volatiles
Authors have presented on the Effect of yoghurt and its components on the deodorization of raw and fried garlic (Allium sativum L.) volatiles.
Garlic contains sulfur volatiles that cause bad odor after consumption, while yoghurt deodorizes these volatiles and
therefore, this has become the objective of this study to understand how the components of yoghurt cause deodorization.
Authors have thoroughly tested yogurt understanding the mechanism behind the deodorization of both raw and fried garlic with yogurt and its components
The manuscript is very informative and with good Food and Nutrition applications.
Yoghurt or yogurt?
Frying garlic significantly reduced the concentration of almost all sulfur volatile compounds.
Yoghurt also significantly reduced the headspace concentration of all major odor-producing garlic volatiles
including allyl methyl disulfide, diallyl disulfide, allyl mercaptan, and allyl methyl sulfide, in both raw and fried garlic.
The components responsible for the deodorization of the volatiles were fat, protein and water.
The title was good
Well written and structured.
Contain adequate tables and several well-drawn figures, equally informative
Literature may be increased slightly
Conclusions are evidenced by data
Elaborate on Results and discussion, may be clear mechanism if any
References can be increased are just adequate, authors may revisit it.
Recheck for English grammar.
Accept the manuscript for publication with minor mandatory changes
With Regards,
Author Response
Yoghurt or yogurt?
The correction is made in the paper. The word Yogurt is used now.
Literature may be increased slightly
We have included the literature that is needed to describe our results.
Elaborate on Results and discussion, may be clear mechanism if any
We added additional discussion in section 2.5
References can be increased are just adequate, authors may revisit it.
We have included the references that are needed to describe our results.
Recheck for English grammar.
Done.
Reviewer 3 Report
This manuscript focused on reduction the unpleasant odor associated with garlic consumption. The manuscript is well-written and the findings are interesting and meaningful.
Author Response
This manuscript focused on reduction the unpleasant odor associated with garlic consumption. The manuscript is well-written and the findings are interesting and meaningful.
Thank you for your comments.
Round 2
Reviewer 1 Report
Thank you for your comments.
Author Response
Thank you for your comments.
Our response:
You are welcome. We have also made some minor edits and additions to the conclusion.